# Betaine and Antioxidants Improve Growth Performance, Breast Muscle Development and Ameliorate Thermoregulatory Responses to Cyclic Heat Exposure in Broiler Chickens

**DOI:** 10.3390/ani8100162

**Published:** 2018-09-25

**Authors:** Majid Shakeri, Jeremy James Cottrell, Stuart Wilkinson, Mitchell Ringuet, John Barton Furness, Frank Rowland Dunshea

**Affiliations:** 1Faculty of Veterinary and Agricultural Sciences, The University of Melbourne, Parkville, Victoria 3010, Australia; jcottrell@unimelb.edu.au (J.J.C.); ringuet.m@unimelb.edu.au (M.R.); j.furness@unimelb.edu.au (J.B.F.); fdunshea@unimelb.edu.au (F.R.D.); 2Feedworks Pty Ltd., Romsey, Victoria 3434, Australia; stuart.wilkinson@feedworks.com.au

**Keywords:** betaine, antioxidant, broiler, growth, heat stress

## Abstract

**Simple Summary:**

The physiological and metabolic responses of broiler chickens to control thermoregulation during heat stress divert energy from efficient production in addition to increasing morbidity and mortality. Therefore, heat stress amelioration strategies may improve the productivity of poultry meat production over the summer months and in tropical regions. Increasingly, low-cost feed additives are being investigated as potential amelioration strategies against heat stress. Scholars have investigated the effects of betaine alone on growth performance and gut physiology, while a small number of studies have been made regarding the impacts of the combination of betaine and antioxidants. Therefore, this study was conducted to investigate the effects of the osmolyte betaine and selenium and vitamin E supplementation on growth performance, physiological responses, and gut physiology in broiler chickens exposed to cyclic heat stress.

**Abstract:**

Heat stress (HS) is an environmental stressor challenging poultry production and requires a strategy to cope with it. A total of 288-day-old male broiler chicks were fed with one of the following diets: basal diet, basal with betaine (BET), or with selenium and vitamin E (AOX), or with a combination of BET and AOX, under thermoneutral and cyclic HS. Results showed that HS reduced average daily feed intake (ADFI) (*p* = 0.01) and average daily gain (ADG) (*p* < 0.001), and impaired feed conversion ratio (FCR) (*p* = 0.03) during rearing period (0–42 day). BET increased ADG (*p* = 0.001) and decreased FCR (*p* = 0.02), whereas AOX had no effects. Breast muscle weight was decreased by HS (*p* < 0.001) and increased by BET (*p* < 0.001). Rectal temperature was increased by HS (*p* < 0.001) and improved by BET overall. Respiration rate was increased by HS (*p* < 0.001), but BET decreased it during HS (*p* = 0.04). Jejunum transepithelial resistance was reduced by HS and had no effect on permeability whereas BET increased jejunum permeability (*p* = 0.013). Overall, the reductions in ADG of broiler chickens during HS were ameliorated by supplementation with BET, with much of the increase in ADG being breast muscle.

## 1. Introduction

Heat stress (HS) causes reduction in feed intake and body weight gain [1], thus posing a major concern for poultry industry. Furthermore, HS increases mortality rate, possibly by impairing intestinal development [2] or disrupting barrier function [3]. It increases radiant heat loss through the redistribution of blood flow from the body core to the periphery, where it can more readily radiate to the environment. This redistribution of blood flow during HS arguably underpins hypoxia and tissue damage within the gastrointestinal tract (GIT) [4]. The GIT forms the digestive and absorptive apparatus of the body, and damages to it is likely to compromise feed efficiency. In addition, reductions in barrier function are likely to inflict a portal of entry for pathogens, increasing disease risk [5,6], and they are of particular concern for intensive production systems with high stocking densities due to the relative ease of transmission of enteric infection.

Betaine (BET) is a natural compound that accumulates within the intestine of broilers when included in diets. The protective effects of dietary BET during HS are threefold. To begin with, BET serves as an extracellular osmolyte, lowering the activity of Na^+^/K^+^ ATPases and therefore reducing overall energy expenditure [7,8]. Because the GIT accounts for a disproportionately large amount of whole-body energy expenditure, the GIT is particularly sensitive to perturbations in blood flow and it has been proposed the reducing the GIT energy demand with BET may serve as a protective mechanism against HS [9]. Furthermore, BET acts as a methyl donor. This has wide-ranging effects, including DNA methylation and increasing methionine remethylation from homocysteine to increase protein synthesis. Finally, osmolytes act as chaperones, stabilising protein folding. In broilers, BET supplementation improves villus morphology, including following coccidia infection [10]. While one study has been inconsistent [11], it has been proposed that BET protects against the effects of HS by ameliorating damage to the GIT [12]. As studies sought to explore the ideal dosage of BET added in broiler chickens’ diets, it has been established that growth performance improved with 0.5–2 g/kg added BET, and within the range of 0.5–1 g/kg, the increase in the improvement is proportional to the increase in the dosage of BET in the diets, with 1 g/kg being the most ideal dosage for such improvement; while BET 1–2 g/kg, albeit effective, did not improve growth performance any further than BET 1 g/kg did [13,14,15].

Selenium (Se) is a trace element that improves growth performance by improving the efficiency of feed utilization [16], by protecting and preserving of cells and cell membrane and is required for expression of the anti-oxidant enzyme glutathione peroxidase (GP_x_). Under thermoneutral (TN) conditions, the dietary Se requirement for poultry is approximately 0.10 to 0.15 mg/kg [17]. However, higher levels of dietary Se, of the order of 0.4 to 1.0 mg/kg, have been recommended to enhance cellular antioxidant levels, improve immunity, and ameliorate the negative impacts of HS [18,19,20]. Furthermore, organic forms of Se, such as selenised yeast or protein-bound Se, may be more available and bioavailable than inorganic forms of Se in broiler chickens [21] and pigs [22].

Vitamin E (Vit E) is a fat-soluble vitamin with anti-oxidant properties [23] that regenerates damaged tissues [24,25] during oxidative stress by participating in the GPx pathway resulting in improved chicken performance [26]. For example, dietary supplementation with up to 250 mg/kg Vit E improved the growth performance of broiler chickens reared under both TN and HS conditions [26,27].

Most of the studies investigating the effects of dietary strategies on alleviating HS have been performed on chickens with only a moderate growth potential, rather than genetically improved chickens. Moreover, based on our best knowledge, a small number of studies have investigated effects of the combination of BET with selenium and vitamin E in broiler chickens of Ross-308 under HS. Therefore, we hypothesize that BET, Se, and Vit E might ameliorate some of the effects of HS and therefore the aims of this experiment were to determine the effects of supplemental BET and antioxidants (AOX), either alone or in combination on growth performance, breast muscle development, and the physiological responses to HS in contemporary broilers chickens.

## 2. Materials and Methods

### 2.1. Ethics Approval and Consent to Participate

The study was performed in Animal facility in The University of Melbourne, Parkville, Victoria in Australia. All procedures of this experiment were approved by the Animal Ethics Committee of the Faculty of Veterinary and Agricultural Sciences, The University of Melbourne, Australia (Protocol no. 1613936.1). The use of animals for research is directed by federal and state legislation according to the Prevention of Cruelty to Animals Act 1986 and the Prevention of Cruelty to Animals Regulations 2008.

### 2.2. Birds and Husbandry Housing 

A total of 288 day-old-male Ross-308 chicks were obtained from a local commercial hatchery (Tri Foods Pty. Ltd., Bannockburn, Victoria, Australia) located within less than 2 h driving distance from The University of Melbourne. Chicks were transported in three boxes (100 chicks per box) in an environmentally controlled vehicle (where the temperature inside the vehicle was ~30 °C) to the experimental site. Chicks were randomly selected and then weighed as a group of six chicks and allocated to equally-sized pens (1 × 0.5 m) with wire flooring and walls and with the floors covered with wood shavings (10 cm deep approximately) in two environmentally controlled rooms (24 pens in each room). Temperature rooms were operated independently and there were temperature and humidity sensors located within each room. For the first seven days of the experiment, the temperature of both rooms was a constant 33 °C for the full 24 h in each day. From day 8, the temperature in each room was either TN where the temperature was decreased gradually from 33 °C to 25 °C on 21 days (one degree every two days starting from day 7) and then maintained at 25 °C from 21–42 days, which is in the range of the optimum temperature for Ross-308 broiler chickens [28] or HS conditions. The HS conditions were characterized as cyclic 33 °C from 1–42 days where the chickens were kept at 33 °C between 08:00 to 17:00 h and then the prevailing temperature in the TN room for the rest of the day. The relative humidity was between 45% and 65% during the experiment. Light (fluorescent, 10 lux) was provided for 24 h for the first three days after placement and gradually decreased (1 h per day) to 20 h at day 7 for both rooms. Each pen was equipped with a nipple drinker and a ball type feeder. To minimize feed wastage during the experiment, each feeder was placed on a box with a net on the top to collect spilled feed. Each room was equipped with two fans to exchange air (~0.2 m^3^/min).

### 2.3. Experimental Treatments

The experiment was a 2 × 2 × 2 factorial design the respective factors being environmental temperature (TEMP) regime (TN versus HS), added dietary BET (0 versus 1 g/kg BET (Betafin S1, DuPont, Marlborough, UK) and dietary AOX (Basal (50 IU/kg Vit E and 0.3 mg Se/kg) and 300 IU/kg Vit E and 0.8 mg Se/kg). The basal AOX diet was formulated to satisfy NRC requirements [17] with Se provided as selenised yeast (SelenoSource 3000, Diamond V Mills, Cedar Rapids, USA) and the Vit E as natural Vit E (Vit E 250, dl-α-tocopheryl acetate, ADM, Chicago, IL, USA). Feed (Feedworks_BESTMIX, The University of Sydney) was formulated as a commercial starter was crumbled from 1–14 days (CP 22.19% ME 2950 kcal/kg), grower was pelleted from 15–28 days (CP 20.54% ME 3100 kcal/kg) and finisher was pelleted from 29–42 days (CP 19.16% ME 3.200 kcal/kg) (Table 1). It has been recommended [29] that the finisher diet should be provided for broiler chickens after approximately 25 days. Feed and water were provided ad libitum and live weight and feed disappearance were recorded weekly with a digital scale (Teraoka Seiko, S-YBS, Tokyo, Japan) to calculate weekly average daily gain (ADG), average daily feed intake (ADFI) and feed conversion ratio (FCR). Mortality rate was recorded daily. All chicks were vaccinated against coccidiosis and Newcastle in the hatchery to avoid coccidiosis during the rearing period.

### 2.4. Physiological Responses

Physiological responses to HS were measured by quantifying rectal temperature and respiration rate. Forty-eight chickens (one per pen) were randomly selected to measure rectal temperature weekly from day 7 at 7:00, 12:00, and 16:00 by using a digital thermometer (Comark Instruments, PDT 300, Norwich, UK) inserted about 3–4 cm into the cloaca for 35 s until the temperature stabilized. Forty-eight chickens were randomly selected to measure respiration rate at 12:00 on day 41. Chickens were filmed using an android mobile phone (Samsung Electronics Co., Ltd., Yeongtong-gu, Korea) and then the number of breaths taken over a 10 s period was quantified and then expressed as breaths per minute. The same researcher measured rectal temperature and filmed and measured respiration rate to keep consistency. 

### 2.5. Tissue Sampling and Euthanasia 

At 21 days of age half of the chickens (three per pen) were randomly chosen to be slaughtered to obtain liver samples for subsequent GPx analysis with the remaining chickens slaughtered at 42 days to measure gut integrity, permeability, breast muscle weight and abdominal fad pat. Abdominal fat pat was collected and weighed after slaughtering. Chickens were killed by injecting 3 mL lethobarb (pentobarbitone sodium, Virbac Animal Health, Milperra, NSW, Australia) into the wing vein; death was confirmed by the absence of heartbeat and respiratory movement, by observation and lack of response to a toe pinch.

### 2.6. Intestinal Barrier Integrity and Permeability

At day 42, jejunum samples were collected from 80 randomly pre-selected chickens. The mucosal integrity and permeability were measured as previously described [30] using chambers by trans-epithelial resistance (TER) and FITC-Dextran 4 kDa (FD4) permeability. In brief, fresh jejunum samples were inserted into chambers (EasyMount Diffusion Chambers; Physiologic Instruments, San Diego, CA, USA). Each chamber had a set of four electrodes and contained 5 mL of the Krebs bicarbonate buffer. Tissues were allowed to equilibrate for about 25 min in the chambers before measurements were made. Transepithelial electrical resistance was calculated by Ohm’s law and multiplied by the exposed area. Macromolecular permeability was quantified by passage of 4 kDa fluorescein isothiocyanate dextran from the mucosal side (1 mg·mL^−1^) to the basolateral side. The FD4 fluorescence was measured using a fluorescence reader (FlexStation II; Molecular Devices, Sunnyvale, CA, USA). The apparent permeability coefficient of FD4 was calculated by equation given by [31].

### 2.7. Glutathione Peroxidase Assay

Glutathione peroxidase was measured kinetically in liver homogenates by the oxidation of NADPH and measured at 340 nm as per the manufacturer’s instructions (Cayman Chemical Company, Ann Arbor, MI, USA). Briefly, non-enzymatic and positive control wells were prepared by diluting 120 µL of assay buffer and 50 µL of co-substrate mixture, and diluting 100 µL of assay buffer, 50 µL of co-substrate mixture, and 20 µL of diluted GPx, respectively. Samples wells were prepared by adding 100 µL of assay buffer, 50 µL co-substrate mixture and 20 µL samples. Then, 20 µL cumene hydroperoxide was added to all wells and gently shook for 10 s. Absorbance was read five times at 340 nm. Then, actual values were calculated by a provided formula based on obtained absorbance results.

### 2.8. Statistical Analysis

All growth performance, breast muscle development, abdominal fat pad, carcass, and respiration rate data were analysed by ANOVA for the main and interactive effects of TEMP, dietary BET or dietary AOX with pen as the experimental unit using the GLM procedure [32]. Mortality data were analysed using the χ^2^ maximum likelihood contingency test to determine the main effects of TEMP, BET, and AOX. Rectal temperature was analysed for main and interactive effects of day of study (DAY), time of day (TM), TEMP, BET, and AOX with pen as the random effect. Means were separated using Duncan’s multiple range test. Significance was considered as *p <* 0.05. 

## 3. Results

### 3.1. Performance

Between days 21 and 42 there was a reduction in ADFI (*p* = 0.001) and ADG (*p* < 0.001) and an increase in FCR (*p* = 0.04) in response to HS (Table 2). Dietary BET increased ADG (*p* = 0.001) and decreased FCR (*p* = 0.04) between days 21 and 42. Although there were no main effects of AOX on either ADFI, ADG, or FCR between days 21 and 42 of the experiment. Over the entire experiment, there was a reduction in ADFI (*p* = 0.01) and ADG (*p* < 0.001) and an increase in FCR (*p* = 0.03) in response to HS (Table 2). Dietary BET increased ADG (*p* = 0.001) and decreased FCR (*p* = 0.02) between days 0 and 42 but had no significant effect on ADFI. Although there were no significant main effects of AOX on either ADFI, ADG, or FCR over the entire experiment, there tended (*p* = 0.09) to be an effect of AOX and an interaction between TEMP and AOX (*p* = 0.02) such that AOX decreased ADFI under TN conditions but not during HS (Table 2). As a result of the effects on ADG, final live weight was decreased by HS (*p* < 0.001) and increased by dietary BET (*p* = 0.003) whereas there was no effect of AOX. A large part of the changes in live weight appeared to be as breast muscle which was decreased by HS (*p* < 0.001) and increased by dietary BET (*p* < 0.001) with no main effect of AOX. However, there were significant TEMP × AOX (*p* = 0.046), AOX × BET (*p* = 0.007) and TEMP × AOX × BET (*p* = 0.015) interactions such that the combined BET and AOX treatment increased ADG during HS but not under TN conditions (Table 2). There were no main effects of HS, BET, or AOX on fat pad weight, although there was a significant AOX × BET (*p* = 0.019) interaction indicating that AOX alone increased fat pad weight, but not when combined with BET. The mortality rate was low but was increased by HS (0.7 versus 4.2%, χ^2^ = 4.05, 1 df, *p* = 0.044) whereas there was no effect of dietary BET (1.4 versus 3.5%, χ^2^ = 1.36, 1 df, *p* = 0.24) or AOX (2.1 versus 2.8%, χ^2^ = 0.15, 1 df, *p* = 0.70). However, these data must be treated with caution because of the low mortality rates and hence low frequency in some cells.

### 3.2. Physiological Responses

Heat stress increased rectal temperature (*p* < 0.001) with the response being greatest (*p* < 0.001) during the afternoon when the ambient temperature was highest (Figure 1b) and no effects under TN (Figure 1a). Although there were no main effects of BET, there was a reduction in rectal temperature in response to dietary BET during the initial stages of the experiment in chickens exposed to HS, particularly late in the afternoon, as indicated by DAY × BET (*p* < 0.001), TM × BET (*p* = 0.015), and TM × DAY × BET (*p* = 0.04) interactions (Figure 1b). Also, the positive effects of BET on rectal temperature were reduced when combined with dietary AOX as indicated by DAY × BET × AOX (*p* < 0.001) and TM × DAY × TEMP × BET × AOX (*p* < 0.014) interactive effects (Figure 1b,d). Interestingly, rectal temperature increased at the end of the experiment in chickens fed the AOX diet and housed under TN conditions as indicated by the DAY × TEMP × AOX (*p* < 0.001) interaction (Figure 1c).

Respiration rate was increased by HS (*p* < 0.001) and decreased by BET (*p* = 0.017) particularly during HS conditions as indicated by the TEMP × BET (*p* = 0.048) interaction (Figure 2). There were no main or interactive effects of AOX on respiration rate (Figure 2).

### 3.3. Intestinal Epithelial Barrier Integrity

Heat stress reduced jejunum TER by approximately 30% compared to their TN counterparts (149 vs. 104 Ω/cm^2^) but there were no main or interactive dietary effects (Figure 3). Heat stress had no effect on jejunum FD4 permeability (*p* = 0.24) whereas dietary BET increased jejunum FD4 permeability (*p* = 0.013) compared to control chickens (Figure 4).

### 3.4. Glutathione Peroxidase Activity

All of the main and interactive effects of HS, day, BET, and AOX on liver GPx activity were significant (*p* < 0.001) (Figure 5) which requires a comprehensive interrogation of the data to obtain a clear picture. There was no effect of HS on GPx in the chickens consuming the control diet on day 21 (12.5 vs. 11.9 nmol/(mg protein min)) whereas GPx on day 42 decreased in those chickens maintained under TN conditions but was unchanged in those exposed to HS (6.33 vs. 13.9 nmol/(mg protein min)). Dietary BET and AOX increased GPx on day 21 but the effects were not additive and there was no main effect of HS at this time. While there was a decrease in GPx activity between day 21 and day 42 for all dietary groups under TN conditions, nevertheless supplemental BET and AOX increased GPx over that of the controls. On day 42, the effects of dietary BET and AOX were exacerbated by HS as indicated by highly significant three- and four-way interactions (Figure 5).

## 4. Discussion

### 4.1. Effects of Betaine, Selenium, and Vitamin E on Growth Performance and Physiological Responses 

Under some rearing conditions HS can be a major cause of impaired performance, reduced muscle development and increased mortality rate in chickens. For example, HS impairs growth performance [33,34] by increasing body temperature and panting rate due to respiratory alkalosis with subsequent growth depression in chickens [35]. During HS, the chicken diverts energy towards reducing their body temperature and away from growth and breast muscle development through different metabolic mechanisms [35]. This experiment confirmed that HS increased rectal temperature (+1.3 °C) and respiration rate (+223%), and reduced ADG (−11%), ADFI (−4%), and breast weight (−16%) in broiler chickens fed a conventional commercial ration. One major finding from the present experiment was that supplementation with 1 g/kg BET partially ameliorated rectal temperature, respiration rate, ADFI, and ADG and fully ameliorated breast muscle weight in chickens exposed to chronic cyclic HS, which is in line with [36,37]. The effects of AOX were not as pronounced although AOX reduced rectal temperature and respiration rate during the hottest part of the day in chickens exposed to HS conditions. The results can be explained by improvements obtained from AOX which are an important part of many metabolic pathways, and show that AOX can enhance the immunity, which decreases body temperature and reduces oxidative damages in cell [38]. Importantly, these improvements were observed in rapidly growing chickens of an improved genotype. Furthermore, dietary BET but not AOX, improved ADG (+7%), FCR (−3%), final live weight (+7%), and breast muscle (+25%) in chickens housed under TN conditions. While others have found that dietary BET can increase ADG and final live weight [36,37,39,40] and reduce rectal temperature and respiration rate [40] during HS, the present experiment is the first with an improved genotype capable of reaching 2.8 kg by 42 days of age under TN conditions. By comparing the production results of different genotypes of broiler chickens [41,42], it can be inferred that breed selection, genetic progress, and nutritional strategies are among the most cost-effective approaches to increasing poultry production. Among different broiler breeds, Ross and Cobb breeds have achieved better final growth performance, and a recent study [43] showed even better performance for Ross-308 under HS. Previous studies [13,36,37,44] observed growth improvement in other genotypes of broiler chickens when different levels of BET were supplemented to the diet, while results of the present study observed the greatest improvement in the growth performance of Ross-308 supplemented with 1 g/kg BET compared to other genotypes under HS. Dietary BET can assist the performance during HS by playing an important role in the methionine–homocysteine cycle by acting as a methyl group donor [45]. Dietary BET may spare methionine from methyl donation for other metabolic mechanisms such as protein synthesis and immune modulation [46]. Moreover, BET alleviates the negative effects of osmotic stress in the intestine and other tissues [47], eliminates toxic metabolites [45], and enhances antioxidant status [13].

### 4.2. Effects of Betaine, Selenium, and Vitamin E on Gut Permeability and Integrity

Heat stress may impair growth performance and health by damaging intestinal barrier function in the gut [48] and the findings from this experiment confirm that HS reduces TER in broilers [49,50,51]. The reason for this phenomenon is likely due to a reduction in blood flow to intestinal and other visceral tissue which can cause hypoxia and resultant oxidative damage. Despite the decrease in TER, HS did not increase permeability compared to TN which is in contrast to [52]. The reason may be related to the positive impacts of BET and AOX under HS conditions. Betaine has been shown to improve growth performance in pigs [52], poultry [43,53], and sheep [54]. These positive effects can be explained by improved intracellular water retention, protection of intracellular enzymes against osmotically induced inactivation [40] and a reduction in energy expenditure. In the present study, dietary BET increased permeability of jejunum compared to other diets. Dietary AOX and BET can reduce gut cell damage and improve (reduce) intestinal permeability in broilers [49,50]. The reason for differences between our results with other studies may be due to the difference in the site of measurement. Betaine is mostly absorbed in the duodenum and has a greater effect on the duodenum because osmotic pressure in this section is higher [47,55].

### 4.3. Effects of Betaine, Selenium, and Vitamin E on Glutathione Peroxidase Activity

Heat stress disturbs the balance between the production of reactive oxygen species (ROS) and GPx activity in broiler chickens [56,57,58]. Increased ROS concentrations lead to oxidative damage and decreased GPx activity [59]. Glutathione peroxidase production and activation is an adoptive mechanism in the cellular anti-oxidant defence system which is increased when free radical production increases to deal with oxidative stress to protect against tissue damage [60]. These results are in concordance with those of [61,62,63] where cyclic HS increased liver GPx activity in chickens. Moreover, Se, vit E, and BET supplementation increased GPx activity in broiler chickens [36,64,65,66] and pigs [30]. The present data indicate that the decrease in GPx levels over time in chickens that were not supplemented with BET or AOX is possibly related to a reduction in GPx synthesis, and that this mechanism can be enhanced by Se that helps regulate the synthesis of GPx enzyme [67]. The reduction in GPx over time may also indicate that as fractional growth rate decreases, so too does the production of ROS and hence the need for GPx. Moreover, BET enhances anti-oxidant defence [47] by acting as a free radical scavenger to neutralize free radicals, and promotes the main antioxidant enzyme activities including GPx [36,68,69].

### 4.4. Betaine, Selenium, and Vitamin E Functions During Heat Stress

The ability of BET to stimulate ADG and breast muscle under TN conditions has also been observed previously [46], although the magnitude of the effects here were impressive given the high basal performance of the control chickens. The breast muscle is the most valuable muscle in the chicken carcass [70], so an increase in muscle weight of the magnitude observed here is profound. When protein deposition is limited by energy intake, the energy spared from the reduction in ion-pumping during BET supplementation can be used to stimulate skeletal muscle deposition [71]. It is likely that the protein deposition potential of rapidly growing chickens of an improved genotype is limited by energy intake [72], therefore dietary BET would be expected to stimulate skeletal muscle deposition. 

Selenium protects tissues against cell damage and excessive ROS generation via the glutathione peroxidase pathway, thereby protecting tissues against oxidation of lipids and proteins [73]. Vitamin E regenerates damaged cells by reducing synthesis of malonyldialdehyde and protecting them against lipid peroxidation and cell damage [74], thereby improving chicken growth performance [26]. In the present experiment, AOX decreased growth performance under TN conditions but not under HS conditions. In addition, the rectal temperature in chickens fed supplemental AOX, particularly when combined with BET, was increased towards the end of the experiment in chickens housed under TN conditions. These findings are consistent with findings of [45,75] who found that supplementation with Se above 0.3 mg/kg impaired growth performance in chickens in tropical climatic conditions. Moreover, Ref. [76,77] indicated that the combination of 0.5 mg/kg Se with 250 IU/kg Vit E or 0.5 mg/kg Se with 200 IU/kg Vit E had no positive impacts on chicken performance. However, in other studies supplementation of up to 0.5 mg/kg Se alone [65] or 250 IU/kg Vit E alone [26] improved performance. Interestingly, Ref. [78] found an interaction between AOX and thermal treatment such that AOX reduced the HS-induced increase in reactive oxygen species whereas under TN conditions AOX tended to increase reactive oxygen species and other indices of oxidative damage. These findings indicate that when there is little oxidative stress, such as under TN conditions, then excess AOX may create pro-oxidant conditions. Despite there not being any growth promoting effects, supplementation with AOX did reduce rectal temperature and respiration rate during HS in the present experiment, indicating some partial mitigation against HS.

## 5. Conclusions

Betaine has an important role as an osmolyte and methyl donor in poultry and therefore may be useful under dysfunctional osmoregulatory conditions such as HS. The results of this experiment demonstrate that supplementation of 1 g/kg BET individually, or in combination with AOX, could alleviate the effects of HS and improve ADG, ADFI, FCR, and breast development of Ross-308 broiler chicken for up to six weeks. However, AOX did not improve growth performance under TN condition which may be related to creation of pro-oxidant conditions. Moreover, the beneficial effects of BET might be related to lower rectal temperature and respiration rate during HS, which helps chickens convert more consumed energy and nutrients into meat rather than mechanisms to reduce body temperature.

## Figures and Tables

**Figure 1 animals-08-00162-f001:**
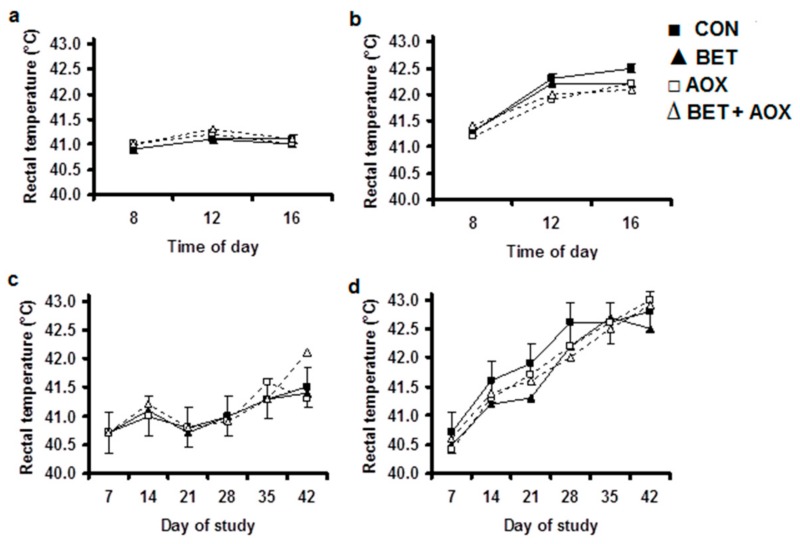
Rectal temperature in chickens fed either a control diet (CON, filled square), the control diet plus betaine (BET, filled tringle), the control diet plus supplemental antioxidants (AOX, open square) or the control diet plus both BET and AOX (open tringle) under either thermoneutral (**a**,**c**) or cyclic heat stress (**b**,**d**). Panels (**a**,**b**) indicate the effect of time of day (TM) pooled across days of the experiment with the standard error of the difference for the interaction between TM, BET and AOX displayed on the data from the chicken receiving the control diet. Panels (**c**,**d**) indicate the effect of day of experiment pooled across time of day with the standard error of the difference for the interaction between day of experiment (DAY), BET and AOX displayed on the data from the chicken receiving the control diet. There were significant main effects of temperature (TEMP) (*p* < 0.001), DAY (*p* < 0.001) and TM (*p* < 0.001) and TEMP × AOX (*p* = 0.048), TM × DAY (*p* < 0.001), TEMP × TM (*p* < 0.001), DAY × TM (*p* < 0.001), TM × BET (*p* = 0.015), DAY × BET (*p* < 0.001), TM × DAY × BET (*p* = 0.040), DAY × TEMP × AOX (*p* < 0.001), DAY × BET × AOX (*p* < 0.001), and TM × DAY × TEMP × BET × AOX (*p* < 0.014) interactive effects. There were no other main or interactive effects (*p* < 0.05).

**Figure 2 animals-08-00162-f002:**
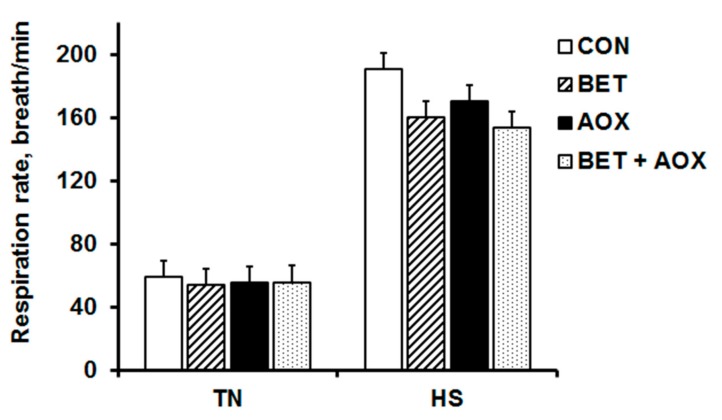
Respiration rate in chickens fed either a control diet (CON), CON plus betaine (BET), CON plus supplemental antioxidants (AOX) or CON plus both BET and AOX under either thermoneutral (TN) or cyclic heat stress (HS) at 12:00 h on day 41 of the experiment. Data are means with the standard error of the difference for the interaction between temperature (TEMP), BET and AOX. The *p*-values for the effects of TEMP, BET, AOX, TEMP × BET, TEMP × AOX, BET × AOX, and TEMP × BET × AOX were <0.001, 0.017, 0.16, 0.048, 0.24, 0.37, and 0.69, respectively.

**Figure 3 animals-08-00162-f003:**
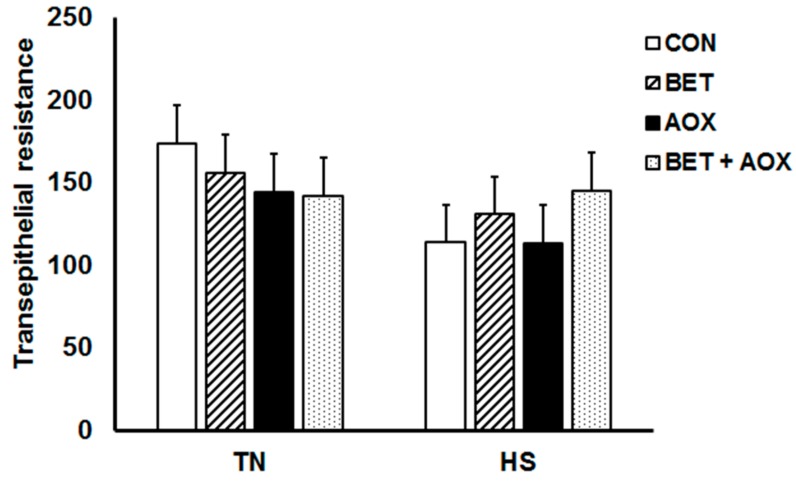
Jejunum transepithelial electrical resistance (TER) in chickens under either thermoneutral (TN) or cyclic heat stress (HS) and fed either a control diet (CON), CON plus betaine (BET), CON plus supplemental antioxidants (AOX), or CON plus both BET and AOX.

**Figure 4 animals-08-00162-f004:**
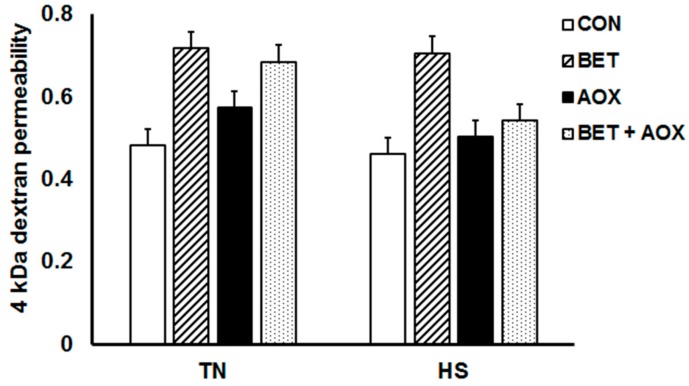
Jejunum fluorescein isothiocyanate (FITC)-dextran (4 kDa) permeability in chickens under either thermoneutral (TN) or cyclic heat stress (HS) and fed either a control diet (CON), CON plus betaine (BET), CON plus supplemental antioxidants (AOX), or CON plus both BET and AOX.

**Figure 5 animals-08-00162-f005:**
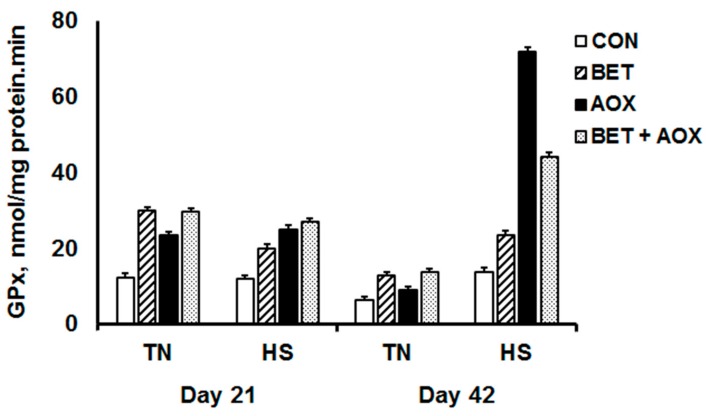
Glutathione peroxidase activity (GPx) in chickens under either thermoneutral (TN) or cyclic heat stress (HS) and fed either a control diet (CON), CON plus betaine (BET), CON plus supplemental antioxidants (AOX), or CON plus both BET and AOX at day 21 and 42.

**Table 1 animals-08-00162-t001:** Nutrient composition of commercial experimental diets

Ingredients %	Starter	Grower	Finisher
Wheat	57.86	60.82	63.61
Soybean meal 48%	31.13	25.28	19.69
Canola meal 38%	4.00	6.00	8.00
Soya oil	3.21	4.55	5.68
Dicalcium phosphate	1.27	0.99	0.81
Limestone	1.12	1.04	0.96
Sodium bicarbonate	0.25	0.25	0.31
Lysine	0.25	0.23	0.20
Celite	0.20	0.20	0.20
DL-methionine	0.28	0.23	0.19
Salt	0.20	0.20	0.16
Starter premix	0.10	0.10	0.10
Threonine	0.11	0.09	0.06
Axtra XB 201 ^†^	0.01	0.01	0.01
Axtra Phy TPT 10,000 ^‡^	0.01	0.01	0.01
Total	100	100	100
Calculate analysis			
AME ^1^ (kcal/kg)	2950.00	3100.00	3200.00
Crude protein (%)	22.19	20.54	19.16
Sodium (%)	0.16	0.16	0.16
Fat (%)	5.04	6.46	7.35
Salt (%)	0.27	0.28	0.24
Calcium (%)	0.96	0.87	0.79
Chloride (%)	0.23	0.23	0.20
Lysine (%)	1.37	1.23	1.10
Methionine (%)	0.61	0.54	0.48
Starch (%)	35.54	37.18	38.90
*p* available	0.48	0.44	2.00

^†^ Contained: endo-1,4-β-xylanase; endo-1,3 (4)-β-glucanase. ^‡^ Contained: dried Trichoderma reesei; sodium sulphate; starch; hydrate magnesium silicate; sucrose; poly-vinyl alcohol; vegetable oil; phytic acid. ^1^ Apparent metabolisable energy.

**Table 2 animals-08-00162-t002:** Effect of heat stress and dietary betaine and antioxidants on growth performance ^1^

Temperature (TEMP)	Thermoneutral				Heat Stress							
Antioxidants ^2^ (AOX)	Control		Supplemental		Control		Supplemental			*p*-Values		
Betaine (g/kg)	0	1.0	0	1.0	0	1.0	0	1.0	SE ^3^	TEMP	AOX	BET
Days and items *												
0 to 21 days												
ADFI (g/d)	59.2	56.7	54.4	60.1	62.0	58.4	57.4	55.3	2.82	0.64	0.12	0.66
ADG (g/d)	36.6	36.3	35.2	35.2	34.1	36.4	35.5	36.3	1.58	0.87	0.57	0.49
FCR	1.62	1.56	1.55	1.71	1.82	1.61	1.62	1.43	0.10	0.52	0.35	0.34
21 to 42 days												
ADFI (g/d)	162	173	167	169	161	155	153	159	6.30	0.001	0.82	0.30
ADG ^4^ (g/d)	93.5	102	91.8	91.4	80.5	86.9	79.6	89.3	3.59	<0.001	0.13	0.001
FCR ^5^	1.74	1.69	1.83	1.86	1.99	1.80	1.93	1.78	0.09	0.04	0.36	0.04
0 to 42 days												
ADFI (g/d)	111	115	111	115	111	109	105	107	3.92	0.01	0.45	0.49
ADG ^6^ (g/d)	65.0	69.3	63.5	63.3	57.6	61.6	57.5	62.8	1.90	<0.001	0.09	0.001
FCR ^7^	1.71	1.66	1.75	1.81	1.93	1.74	1.83	1.70	0.06	0.03	0.60	0.02
Final weight (g)	2769	2949	2704	2699	2457	2627	2454	2675	161.9	<0.001	0.20	0.003
Breast weight ^8^(g)	598	746	612	584	500	587	498	575	66.2	<0.001	0.17	<0.001
Fat weight ^9^ (g)	31.9	38.3	42.7	35.2	28.9	34.0	37.2	33.7	9.31	0.13	0.10	0.95

* ADFI: average daily feed intake; ADG: average daily weight gain; FCR: feed conversion ratio. ^1^ There were no interactions (*p* > 0.10) except where indicated. ^2^ Control diet contained 50 IU/kg Vit E and 0.3 mg Se/kg and 250 IU/kg Vit E and 0.5 mg Se/kg added to control diet for supplemented diets. ^3^ Standard error of the difference for effect of TEMP × AOX × BET. ^4^ Trend for TEMP × AOX (*p* = 0.057) and TEMP × AOX × BET (*p* = 0.088) interactions. ^5^ Trend for TEMP × AOX (*p* = 0.080) and TEMP × BET (*p* = 0.082) interactions. ^6^ Significant TEMP × AOX (*p* = 0.029) interaction. ^7^ Significant TEMP × AOX (*p* = 0.016) and TEMP × BET (*p* = 0.013) interactions. ^8^ Breast weight (m. pectoralis major plus m. pectoralis minor) in broiler chickens aged 42 d. Significant TEMP × AOX (*p* = 0.046), AOX × B (*p* = 0.007) and TEMP × AOX × BET (*p* = 0.015) interactions. ^9^ Abdominal fat pad in broiler chickens aged 42 d. Significant AOX × BET (*p* = 0.019) interaction.

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
