# Peer review of "Betaine and Antioxidants Improve Growth Performance, Breast Muscle Development and Ameliorate Thermoregulatory Responses to Cyclic Heat Exposure in Broiler Chickens"

_animals, 2018, doi:10.3390/ani8100162_

Round 1

Reviewer 1 Report

The authors have studied the effects of betaine and two antioxidants on growth performance and physiological parameters in male chickens exposed to elevated temperature. Betaine was found to improve growth performance without a consistent or major ameliorating effect on elevated temperature. The antioxidants did not provide consistent beneficial effects on growth performance. The study lacks major novelty except that it was undertaken in a genotype with greater potential for growth. The authors need to provide a stronger rationale for why this is of scientific and biological importance. The authors then need to relate the findings in the present study to those of previous studies within the context of differences in growth potential. The manuscript indicates 288 day old chicks which is a remarkable lack of attention to detail. The Abstract states that betaine had no effect on rectal temperature; the Discussion states that betaine partially ameliorated rectal temperature; and the Conclusions states that the benefits of betaine are due to lower rectal temperature. This is imprecise science.  

Author Response

Point 1: The authors have studied the effects of betaine and two antioxidants on growth performance and physiological parameters in male chickens exposed to elevated temperature. Betaine was found to improve growth performance without a consistent or major ameliorating effect on elevated temperature. The antioxidants did not provide consistent beneficial effects on growth performance. The study lacks major novelty except that it was undertaken in a genotype with greater potential for growth. The authors need to provide a stronger rationale for why this is of scientific and biological importance. The authors then need to relate the findings in the present study to those of previous studies within the context of differences in growth potential.

Response 1: L 90 to 92, 305, 312, 314 to 316, 322 to 330, 597 to 599, 600 and 601, and 606 to 608. The following lines have been added.

L 90 to 92.

Moreover, based on our best knowledge, a small number of studies have investigated effects of the combination of BET with selenium and vitamin E in broiler chickens of Ross-308 under HS.

L 305.

[30, 31, 74].

L 312.

which is in line with [31, 33].

L 314 to 316.

The results can be explained by improvements obtained from AOX which are an important part of many metabolic pathways, and show that AOX can enhance the immunity, which decreases body temperature and reduces oxidative damages in cell [75].

L 322 to 330.

By comparing the production results of different genotypes of broiler chickens [77, 78], it can be inferred that breed selection, genetic progress and nutritional strategies are among the most cost-effective approaches to increasing poultry production. Among different broiler breeds, Ross and Cobb breeds have achieved better final growth performance, and a recent study [79] showed even better performance for Ross-308 under HS. Previous studies [31, 32, 33, 38, 76] observed growth improvement in other genotypes of broiler chickens when different levels of BET were supplemented to the diet, while results of the present study observed the greatest improvement in the growth performance of Ross-308 supplemented with 1 g/kg BET compared to other genotypes under HS.

L 597 to 599.  

Hassan R.A.; Attia Y.A.; El-Ganzory E.H. Growth, carcass quality and serum constituents of slow growing chicks as affected by betaine addition to diets containing 1. Different levels of choline. Int. J. Poult. Sci. 2005, 4, 840-850.

L 600 and 601.  

Swan J.; Boles J. Processing characteristics of beef roasts made from high and normal pH bull inside rounds. Meat science. 2002, 62, 399-403.

L 606 to 608.

Dal Bosco A.; Simona M.; Silvia R.; Cecilia M.; Cesare C. Effect of slaughtering age in different commercial chicken genotypes reared according to the organic system: 1. Welfare, carcass and meat traits. Ital. J. Anim. Sci. 2014, 13, 467-472.

Point 2: The manuscript indicates 288 day old chicks which is a remarkable lack of attention to detail. The Abstract states that betaine had no effect on rectal temperature; the Discussion states that betaine partially ameliorated rectal temperature; and the Conclusions states that the benefits of betaine are due to lower rectal temperature. This is imprecise science.  

Response 2: L 40.

From

while unaffected

To

and improved

Reviewer 2 Report

Comments for manuscript ID: animals-animals-351481
The manuscript reports a study evaluating dietary supplementation of Selenium, Vitamin E and Betaine on metabolic and thermoregulatory responses to heat stress in broiler chickens up to 42 day of age. The manuscript does clearly fall within the scope of the journal and does provide some interesting data. Overall, the manuscript is well-written and the study findings will be of interest to the journal readers. However, there are a few areas where revisions/additional information is required (see comments below) and errors that need correcting before the article can be accepted for publication. I consider these revisions to be minor in nature and as such, they should easily be addressed by the authors.
 L27: This is a scientific manuscript so replace ‘meagre’ with a more scientific/accurate term – e.g. ‘a small number of studies’
 L33: Don’t begin a sentence with a number
 Introduction needs a little more to justify Betaine rate of inclusion. Need a better rationale for why authors have included Betaine at 1g/kg? It is assumed this ROI is similar to other studies (references?) but need to state this.
 L68: ‘some’ studies implies more than one but only one citation (Matthews, 2000) has been given here.
 Methods – in what form were diets fed during the phases (meal, pellet?)
 With 6 birds per pen and a ball-type feeder on wood shavings, performance data could have been influenced by wasted feed. How were spillages accounted for during the study to ensure accurate (ADFI and FCR) data?
 L137 to 139: To reduce subjectivity of assessment, need to state who conducted respiration rate evaluation – same individual during the study or a number of people recording respiration rate on phone?
 Performance results: You have set your significance level at P<0.05 so this should be respected. Accordingly, any discussion of tendencies in the dataset outside this level (L183 to 186) should be removed. Same comment applies for L189 to L191. There was no effect of AOX on the growth performance.
 Figures 1a to 1d: Figures and Figure 1 title need to be carefully checked – there is reference to sheep in the title? E.g. L236: ‘data from the sheep’ and L239: ‘AOX displayed on the data from the sheep receiving the control diet’??? This is a study with broilers so either this has been cut/pasted from a different study or there is an error in transcription – either way, this needs to be addressed.

Author Response

Dear Reviewer,

We would like to thank you for the valuable and useful comments given with regards to our manuscript. 

We have made the following corrections/modifications and additions to our manuscript. We also addressed the specific comments and responses are listed out.

Best regards,

Authors

Point 1: L27: This is a scientific manuscript so replace ‘meagre’ with a more scientific/accurate term – e.g. ‘a small number of studies’

Response 1: L27. The following changes have been made.

From

A meagre number of studies

To

a small number of studies’

Point2: L33: Don’t begin a sentence with a number

Response 2: L33. It has been corrected. All the manuscript has been checked for consistency.

From

288-day-old male broiler chicks

To

A total of 288-day-old male broiler chicks

Point 3: Introduction needs a little more to justify Betaine rate of inclusion. Need a better rationale for why authors have included Betaine at 1g/kg? It is assumed this ROI is similar to other studies (references?) but need to state this.

Response 3: L 69 to 74 and 590 to 592. The following lines have been added.

L 69 to 74.

As studies sought to explore the ideal dosage of BET added in broiler chickens’ diets, it has been established that growth performance improved with 0.5-2 g/kg added BET, and within the range of 0.5-1 g/kg, the increase in the improvement is proportional to the increase in the dosage of BET in the diets, with 1 g/kg being the most ideal dosage for such improvement; while BET 1-2 g/kg, albeit effective, did not improve growth performance any further than BET 1 g/kg did [38, 44, 71].

L 590 to 592.

Hassan R.A.; Attia Y.A.; El-Ganzory E.H. Growth, carcass quality and serum constituents of slow growing chicks as affected by betaine addition to diets containing 1. Different levels of choline. Int. J. Poult. Sci. 2005, 4, 840-850.

Point 4: L68: ‘some’ studies implies more than one but only one citation (Matthews, 2000) has been given here.

Response 4: L68. It has been corrected as below.

From

Some studies have been

To

one study has been

Point 5: Methods – in what form were diets fed during the phases (meal, pellet?)

Response 5: L 133 and 134. The forms of the diets have been added.

From

starter from 1-14d

grower from 15-28d  

finisher from 29-42d  

To

starter was crumbled from 1-14d

grower was pelleted from 15-28d

finisher was pelleted from 29-42d

Point 6: With 6 birds per pen and a ball-type feeder on wood shavings, performance data could have been influenced by wasted feed. How were spillages accounted for during the study to ensure accurate (ADFI and FCR) data?

Response 6: L 122 to 124. The following lines have been added.

To minimize feed wastage during the experiment, each feeder was placed on a box with a net on the top to collect spilled feed.

Point 7: L137 to 139: To reduce subjectivity of assessment, need to state who conducted respiration rate evaluation – same individual during the study or a number of people recording respiration rate on phone?

Response 7: L 154 and 155. The following lines have been added.

The same researcher measured rectal temperature, and filmed and measured respiration rate to keep consistency.

Point 8: Performance results: You have set your significance level at P<0.05 so this should be respected. Accordingly, any discussion of tendencies in the dataset outside this level (L183 to 186) should be removed. Same comment applies for L189 to L191. There was no effect of AOX on the growth performance.

Response 8: L 196 and 197, 199 and 200, and 201 to 203. The following lines of the discussion have been removed as recommended.

L 196 and 197.

There was no effect of TEMP, dietary BET or AOX on either ADFI, ADG or FCR between days 0 and 21 of the experiment (Table 2). However,

L 199 and 200.

had no significant effect on ADFI.

L 201 and 203.

there tended to be interactions between TEMP and AOX (P = 0.057) and TEMP, BET and AOX (P = 0.088) such that the combined BET and AOX treatment increased ADG during HS but not under TN conditions (Table 2).

Point 9: Figures 1a to 1d: Figures and Figure 1 title need to be carefully checked – there is reference to sheep in the title? E.g. L236: ‘data from the sheep’ and L239: ‘AOX displayed on the data from the sheep receiving the control diet’??? This is a study with broilers so either this has been cut/pasted from a different study or there is an error in transcription – either way, this needs to be addressed.

Response 8: L 254 and 257. It has been corrected.

From

Sheep

To

Chicken

Reviewer 3 Report

The manuscript contains information on the effect of betaine and anti- oxidants on performance, breast and abdominal fat pad weight and ameliorate thermoregulatory in Ross 308 broiler chickens subjected to heat stress. Manuscript is innovative and written according instruction for authors. Before publication, the manuscript has to be revises and supplemented. Below is the list of proposed changes:

Line 37 "after (P = 0.03)" please add "during rearing period (0-42 d)"

Line 75 [15-17] or [15,16,17]?

Line 101  about 30 ° C is the temperature of the cars box or in the chicks area?

Line 109  25 °C  from 21-42 d of rearing is agree with the recommendations for Ross 308 broiler?

Line 114+, please provide information on the type (incandescent, fluorescent, other?) and intensity of light, air exchange (eg in m3 / h / kg BW)

Line 122-124 CP and ME data are different than in Table 1

Line 123 Why the finisher feed mixture has been used for the last 14 days for rearing broiler chickens, usually is used only 5-7 days before slaughter. This feed mixture doesn’t contain coccidiostats and feed antibiotics (if they are authorized for use in a given country) about the potential negative impact on chicken meat consuments.

Line 124 - please specify the type of balance, manufacturer's data (name, city, country), used to measurement of BW and Feed intake

Table 1 "Soya oil" instead of Soy oil

The "second" word in the name of the component from a lowercase letter

Line 143 - no information about evaluation “Abdominal fat pad”

Line 170 T This manuscript does not contain information about only about breast weight  and abdominal fat pad weight

Line 213 Figures 1 b and 1d instead of Figures 1b and d?

Tables 2 "SE" instead of sed?

No explanation for “breast weight” eg breast weight (m. pectoralis major plus m. pectoralis minor) broiler chickens aged 42 d

Line 228 Abdominal fat pad in broiler chickens aged 42 d

Line 508 "Biochem." instead of "Bioche"

Author Response

Dear Reviewer,

We would like to thank you for the valuable and useful comments given with regards to our manuscript. 

We have made the following corrections/modifications and additions to our manuscript. We also addressed the specific comments and responses are listed out.

Best regards,

Authors

Point 1: Line 37 "after (P = 0.03)" please add "during rearing period (0-42 d)"

Response 1: L 37. It has been added as recommended.

Point 2: Line 75 [15-17] or [15,16,17]?

Response 2: L 80. It has been corrected as [16, 17].

Point 3: Line 101 about 30 ° C is the temperature of the cars box or in the chicks area?

Response 3: L 108. More information has been added.

From

environmentally controlled vehicle (̴ 30°C) to the experimental site.

To

environmentally controlled vehicle (where the temperature inside the vehicle was ̴ 30°C) to the experimental site.

Point 4: Line 109  25 °C  from 21-42 d of rearing is agree with the recommendations for Ross 308 broiler?

Response 4: L 116 and 117, 593 to 595. The following lines have been added.

L 116 and 117.

which is in the range of the optimum temperature for Ross-308 broiler chickens [72].

L 593 to 595.

Kamely M.; Karimi Torshizi M.A.; Rahimi S. Incidence of ascites syndrome and related hematological response in short-term feed-restricted broilers raised at low ambient temperature. Poult. sci2015, 94, 2247-2256.

Point 5: Line 114+, please provide information on the type (incandescent, fluorescent, other?) and intensity of light, air exchange (eg in m3 / h / kg BW)

Response 5:  L 120 and 124. More information has been added.

L 120.

Light (fluorescent, 10 lux).

L 124.

Each room was equipped with two fans to exchange air ( ̴ 0.2 m3/min)

Point 6: Line 122-124 CP and ME data are different than in Table 1

Response 6: L 133 to 135. All the data in the text have been corrected.

From

from 1-14d (CP 22.2% ME 2698 kcal/kg), grower was pelleted from 15-28d (CP 20.5% ME 2677 kcal/kg) and finisher was pelleted from 29-42d (CP 19.2% ME 3.200 kcal/kg)

To

from 1-14d (CP 22.19% ME 2950 kcal/kg), grower was pelleted from 15-28d (CP 20.54% ME 3100 kcal/kg) and finisher was pelleted from 29-42d (CP 19.16% ME 3.200 kcal/kg)

Point 7: Line 123 Why the finisher feed mixture has been used for the last 14 days for rearing broiler chickens, usually is used only 5-7 days before slaughter. This feed mixture doesn’t contain coccidiostats and feed antibiotics (if they are authorized for use in a given country) about the potential negative impact on chicken meat consuments.

Response 7: L 135 and 136, 139 to 141, and 596. The following lines have been added.

L 135 and 136.

It has been recommended [73] that the finisher diet should be provided for broiler chickens after approximately 25 days.

L 139 to 141.

All chicks were vaccinated against coccidiosis and Newcastle in the hatchery to avoid coccidiosis during the rearing period.

L 596.

Aviagen. Ross broiler management handbook. 2014, 11-53.

Point 8: Line 124 - please specify the type of balance, manufacturer's data (name, city, country), used to measurement of BW and Feed intake

Response 8: L 137 and 138. The following line has been added.

with a digital scale (Teraoka Seiko, S-YBS, Tokyo, Japan)

Point 9: Table 1 "Soya oil" instead of Soy oil. The "second" word in the name of the component from a lowercase letter

Response 9: L 142, Table 1. It has been corrected.

Point 10: Line 143 - no information about evaluation “Abdominal fat pad

Response 10: L 159 and 160. The following lines have been added.

abdominal fad pat. Abdominal fat pat was collected and weighed after slaughtering.

Point 11: Line 170 T This manuscript does not contain information about only about breast weight and abdominal fat pad weight

Response 11: L 187. Information about breast muscle and abdominal fat pad have been added.

From

All growth performance, carcass, and respiration rate data were analysed by ANOVA

To

All growth performance, breast muscle development, abdominal fat pad, carcass, and respiration rate data were analysed by ANOVA

Point 12: Line 213 Figures 1 b and 1d instead of Figures 1b and d?

Response 12: L 230. It has been corrected.

Point 13: Tables 2 "SE" instead of sed? No explanation for “breast weight” eg breast weight (m. pectoralis major plus m. pectoralis minor) broiler chickens aged 42 d

Response 13: L 234 Table 2 and 244. sed changed to SE and more changes have been made.

From

Significant TEMP × AOX (P = 0.046), AOX × B (P = 0.007) and TEMP × AOX × BET (P = 0.015) interactions.

To

Breast weight (m. pectoralis major plus m. pectoralis minor) in broiler chickens aged 42d. Significant TEMP × AOX (P = 0.046), AOX × B (P = 0.007) and TEMP × AOX × BET (P = 0.015) interactions.

Point 14: Line 228 Abdominal fat pad in broiler chickens aged 42 d

Response 14: L 246. It has been corrected as recommended.

From

Abdominal fat pad

To

Abdominal fat pad in broiler chickens aged 42 d. Significant AOX × BET (P = 0.019) interaction.

Point 15: Line 508 "Biochem." instead of "Bioche"

Response 15: L 537. It has been corrected.